# Interleukin-6 as a Director of Immunological Events and Tissue Regenerative Capacity in Hemodialyzed Diabetes Patients

**DOI:** 10.3390/medsci12020031

**Published:** 2024-06-15

**Authors:** Maria-Florina Trandafir, Octavian Savu, Daniela Pasarica, Coralia Bleotu, Mihaela Gheorghiu

**Affiliations:** 1Pathophysiology and Immunology Department, “Carol Davila” University of Medicine and Pharmacy, 020021 Bucharest, Romania; daniela.pasarica@umfcd.ro (D.P.); mihaela.gheorghiu@umfcd.ro (M.G.); 2Doctoral School, “Carol Davila” University of Medicine and Pharmacy, 020021 Bucharest, Romania; octavian.savu@umfcd.ro; 3“N.C. Paulescu” National Institute of Diabetes, Nutrition and Metabolic Diseases, 020475 Bucharest, Romania; 4“Stefan S. Nicolau” Institute of Virology, 030304 Bucharest, Romania; cbelotu@yahoo.com

**Keywords:** interleukin 6, interleukin-soluble receptor, neurotrophin-3, vascular endothelial growth factor β, hemodialysis, diabetes mellitus

## Abstract

Hemodialyzed patients have innate immunity activation and adaptive immunity senescence. Diabetes mellitus is a frequent cause for chronic kidney disease and systemic inflammation. We studied the immunological pattern (innate and acquired immunity) and the tissular regeneration capacity in two groups of hemodialyzed patients: one comprised of diabetics and the other of non-diabetics. For inflammation, the following serum markers were determined: interleukin 6 (IL-6), interleukin 1β (IL-1β), tumoral necrosis factor α (TNF-α), IL-6 soluble receptor (sIL-6R), NGAL (human neutrophil gelatinase-associated lipocalin), and interleukin 10 (IL-10). Serum tumoral necrosis factor β (TNF-β) was determined as a cellular immune response marker. Tissue regeneration capacity was studied using neurotrophin-3 (NT-3) and vascular endothelial growth factor β (VEGF-β) serum levels. The results showed important IL-6 and sIL-6R increases in both groups, especially in the diabetic patient group. IL-6 generates trans-signaling at the cellular level through sIL-6R, with proinflammatory and anti-regenerative effects, confirmed through a significant reduction in NT-3 and VEGF-β. Our results suggest that the high serum level of IL-6 significantly influences IL-1β, TNF-β, NT-3, VEGF-β, and IL-10 behavior. Our study is the first that we know of that investigates NT-3 in this patient category. Moreover, we investigated VEGF-β and TNF-β serum behavior, whereas most of the existing data cover only VEGF-α and TNF-α in hemodialyzed patients.

## 1. Introduction

Recent statistical data, from 2023, shows that there are approximately 850 million chronic kidney disease (CKD) cases worldwide [1], 2.7 million of which require hemodialysis. Estimates indicate that in 2030, the number of patients undergoing hemodialysis will reach 5.4 million [2]. According to the Romanian Renal Registry published in 2023, there were 200 dialyzed patients per 1 million inhabitants. As Bucharest has around 2 million inhabitants, the number of dialyzed patients in this city would be around 400. Besides the underlying disease that causes the necessity for chronic hemodialysis, dialyzed patients also suffer from systemic stress, which increases the risk of contracting cardiovascular, neoplastic, and infectious diseases [3]. The basis of these pathologies, the underlying predisposition, is represented by innate immunity activation (inflammation), as well as adaptive immunity senescence (humoral and cellular immune responses). We must also mention the frequent association with diabetes mellitus as a background disease, which is itself a cause of chronic kidney disease and induces chronic systemic inflammation. For these reasons, there is a new concept being used for hemodialyzed patients—“inflamm-aging”—that represents a persistent, low-intensity, sterile inflammatory status that is associated with immune system senescence [4,5], manifested through the shortening of lymphocyte telomer length [6].

Overproduction of proinflammatory cytokines (such as IL1, IL6, or TNF alpha) and reactive oxygen species in patients with chronic kidney disease is mediated by the activation of monocytes and macrophages [7,8]. Furthermore, chronic inflammation in chronic kidney disease augments and is augmented by anorexia and depression, both symptoms recognized as clinical hallmarks of chronic kidney disease [9]. Nevertheless, cardiovascular adverse events in patients with chronic kidney disease are consistently predicted by proinflammatory cytokines, especially IL-6 [10]. Those serological markers of inflammation are also the best predictors for declines in glomerular filtration rate and increases in albuminuria in chronic kidney disease [11].

Activated macrophages and monocytes also seem to be the basis for the overproduction of proinflammatory cytokines in diabetic chronic kidney disease. However, how these markers interact with the progression of chronic kidney disease in diabetes is still controversial [12]. Apart from classical inflammatory cytokines (such as TNF, IL1, or IL6), new markers of inflammation have arisen as important pathogenic mediators for the progression of chronic kidney disease in diabetes (i.e., IL-1 beta, caspase 1-dependent NLRP3) [13].

However, in patients undergoing renal replacement therapy (either hemodialysis or continuous peritoneal dialysis), irrespective of the presence of diabetes mellitus, the serum levels of inflammatory markers (i.e., IL-6, IL beta, IL 10, or TNF alpha) are heterogenous [14,15,16]. Nonetheless, a higher level of inflammation increases the incidence of cardiovascular events in end-stage renal disease; see also [17].

Very recent data attest to the role of neurotrophin-3 (NT-3) in cardiovascular disease. Hence, it seems to reduce the inflammatory and immune response in patients with endothelial dysfunction [18] or with clinically manifest cardiovascular events [19,20]. Moreover, neurotrophins seem to be modulators of inflammation in diabetic patients with established chronic complications, such as retinopathy and diabetic neuropathy [21,22]. However, very few data are available regarding the significance of neurotrophins in the onset and progression of chronic kidney disease [23].

Therefore, we designed a pilot study aiming to analyze the humoral and cellular immunological pattern (innate and acquired immunity) and the tissular regeneration capacity in the serum of two groups of hemodialyzed patients, with and without diabetes mellitus.

## 2. Patients

The present study is an observational, case–control, cross-sectional type of research.

Two groups of hemodialyzed patients (n = 83) were included: a group of cases consisting of 21 individuals with diabetes and end-stage renal disease (DM group) (9 men and 12 women, with an average age of 63 ± 11.8 years) and a group of controls, consisting of 62 hemodialyzed patients without diabetes (NON-DM group) (33 men and 28 women, with an average age of 61.5 ± 15.1 years). The ratio between the DM and NON-DM groups was 1 to 3. The duration of chronic kidney disease was 8.26 ± 6.99 years in patients with diabetes on hemodialysis, and 5.23 ± 4.34 years in subjects on hemodialysis without diabetes. The average time on hemodialysis (three times per week) was 5.29 ± 4.71 years in subjects without diabetes (NON-DM), and 2.53 ± 2.69 years in those with diabetes (DM). The patients’ characteristics are summarized in Table 1.

All participants gave their informed consent upon inclusion to this study. This study was conducted in accordance with the Declaration of Helsinki and was approved through ethical permit no. 295/17.01.2023.

Inclusion criteria for participants in the present study were >18 years of age; chronic kidney disease on chronic replacement therapy (hemodialysis); and established type 2 diabetes mellitus (for subjects with diabetes). Exclusion criteria for participants in this study were acute inflammatory and infectious diseases; current immunosuppressing treatment; malignancies; acute organ failure; acute vascular pathology; hemoglobinopathy or anemia of other causes than CKD; blood transfusion in the last 3 months; and psychiatric conditions or impaired judgement. T2DM was defined according to the American Diabetes Association criteria.

## 3. Methods

Venous blood samples were collected following overnight fasting, aliquoted, stored at −80 °C, and thawed once before further analysis.

All the aforementioned compounds were determined from serum samples, utilizing Merck Millipore ELISA kits for IL-6, IL-1β, IL-10, NT-3, and VEGF β, and Elabscience ELISA kits for IL-6R, TNFα, TNFβ, and NGAL.

The human ELISA kits utilized in our study were single-wash 90 min sandwich ELISAs designed for the quantitative measurement in human serum, plasma, and cell culture supernatant. SimpleStep ELISA^®^ technology employs capture antibodies conjugated to an affinity tag that is recognized by the monoclonal antibody used to coat the SimpleStep ELISA^®^ plates. This approach to sandwich ELISA allows for the formation of the antibody–analyte sandwich complex in a single step, significantly reducing assay time.

### Statistical Analysis

Data were processed using the SPSS IBM V26 statistical package, with a statistical significance defined as *p* ≤ 0.05, conventionally adopted for checking the degree of data compatibility with the null hypothesis. The statistical protocol was applied according to the design of a cross-sectional study and the type of data collected. Descriptive statistics, such as mean ± standard deviation, median, and range, were used for quantitative variable presentation (such as age, duration of hemodialysis, and serum markers). For comparison between groups (DM patients vs. NON-DM patients, IL-6, sIL-6R, etc.), an “Independent–Samples t Test or nonparametric tests (independent samples Mann–Whitney U test, Kruskal–Wallis test) were applied after data normality distribution was verified. Qualitative variables (such as gender, and presence or absence of DM) were described as structure or intensity indicators (absolute or relative frequencies). Chi square-X2 or Fisher’s exact test were also applied to examine the association between the categorial variables. Correlation analysis was also performed. Boxplot graphics (of the minimum, first quartile, median, third quartile, maximum, and outliers of the dataset’s parameters) were drawn to show the data series distribution for the global sample or stratified in subgroups of interest.

## 4. Results

There were no confounding factors between groups; there were no significant differences (*p* > 0.05) in terms of age, gender distribution, duration of hemodialysis, or comorbidities (Table 1).

An initial evaluation of the measurements performed showcases an extremely high level of inflammation in hemodialyzed patients, especially the serum concentration of IL-6 (Figure 1)

All measurements performed using the same type of ELISA kit in individuals without acute inflammatory or infectious disease revealed values under 50 pg/mL. Serum levels of IL-6 were high in both subgroups of subjects, especially in those with diabetes when compared with those without diabetes (median 518.5 pg/mL, range 0–4966.93 pg/mL vs. median 141.84 pg/mL, range 0–2145.17 pg/mL; *p* = 0.05). The same distribution was observed for IL-6R serum levels in subjects with diabetes (median 1511.75 ng/mL, range 416.75–1849.25 ng/mL) when compared with the non-diabetics (median 1220.5 ng/mL, range 611.75–1886.75 ng/mL; *p* = 0.046), (Figure 1).

The value distribution for both serum IL-6 and IL-6R was heterogenous in diabetic subjects while relatively homogenous in non-diabetics (Figure 2).

We observed a significant negative correlation between IL-6 and IL-6R serum values in diabetic subjects only (*p* < 0.012; r^2^ = −0.63).

In order to identify the effects induced by the very high serum levels of IL-6, we restructured the initial group into four subgroups/categories (Table 2), according to the ratio between the serum concentration of IL-6 obtained and the upper limit of the normal range, as follows:-Regular IL-6 (under 50 pg/mL) = ILN (2 patients with diabetes and 13 without);-IL-6 increased by 2–10 times the regular value (100–500 pg/mL) = IL 2–10; (4 patients with diabetes and 19 without);-IL-6 increased by 10–20 times the regular value (500–1000 pg/mL) = IL 10–20 (7 patients with diabetes and 11 without);-IL-6 increased by over 20 times the regular value, so a minimum of 21 times (over 1000 pg/mL) = IL 21 (7 patients with diabetes and 13 without).

The rest of the patients (one patient with diabetes and six without) had increased IL-6 serum levels, but less than two times the regular value (50–100 pg/mL).

We consecutively detected similar concentrations of both IL-6 and IL-6R irrespective of gender and presence of DM at serum values of IL-6 ranging between 10 and 20 pg/mL (Figure 3). Therefore, this result suggests a differentiation in subgroup behavior between IL-6 and IL-6R.

An interesting aspect was observed for all 83 patients, regardless of the presence of diabetes mellitus. Hence, many of the determined parameters (IL-1β, TNF-β, NT3, VEGF-β, and IL10) showed significantly different serum concentrations in relation to the serum level of Il-6: (*p* = 0.001, *p* = 0.022, *p* = 0.048, *p* = 0.029, and *p* = 0.001, respectively).

Furthermore, the stratified parameter analysis, in relation to the IL-6 subgroups and also the presence or absence of diabetes mellitus, revealed the following:-In hemodialyzed diabetes patients, there were no statistically significant differences in parameters relating to general IL-6 determination;-In hemodialyzed patients without diabetes, we obtained statistically significant differences (*p* = 0.05) in parameters relating to the general IL-6 level, as follows: IL1-β vs. IL-6, *p* = 0.034 (Figure 4); TNF-β vs. IL6, *p* = 0.036 (Figure 5); NT-3 vs. IL-6 *p* = 0.046 (Figure 6); VEGFβ vs. IL-6 *p* = 0.034 (Figure 7); IL-10 vs. IL6 *p* = 0.018 (Figure 8).

Thus, all these data show that the high level of IL-6 influences the behavior of IL-1β, TNF-β, NT-3, VEGF-β, and IL-10 in serum of patients on hemodialysis.

There were no significant differences in TNF-α, NGAL, and IL-10 concentrations between the two groups of hemodialyzed patients, nor any significant correlations between serum levels of TNF-α and TNF-β. Further, there were no significant correlations between the serum concentration of IL-6 and those of TNFα and TNFβ (Figure 9 and Figure 10).

However, we observed that serum concentration of NT-3 increases with the concentration of IL-6 and IL-6R only in diabetic patients on hemodialysis (in series, low, high) (Figure 11).

## 5. Discussion

Our results show the presence of inflammation in patients on hemodialysis. We also reported a significantly higher concentration of inflammatory markers in serum of patients with diabetes when compared with those without diabetes.

Our observation is in concordance with data published in the literature that confirm the occurrence of inflammation in subjects on hemodialysis [24] (*Cell Mol Immunol.* 2006;3:151–154).

What are the known mechanisms in generating this inflammation?

During hemodialysis, a series of rapid hemodynamic alterations occur, generating stress (fast ultrafiltration, quick variations in serum electrolyte concentrations, rapid weight loss) [25]. Moreover, the patient’s blood loses the endothelial cells’ protection, which implies antithrombotic action, vascular smooth muscle cell control, thermogenesis, and thermolysis control, allowing for a bidirectional flux of fluids and electrolytes between the blood and tissues and also blocking toxins and other pathogenic compounds from passing into the tissues. To this we add the blood’s contact with the extracorporeal circuit’s components (the needle used for venipuncture; the tubing, blood pumps, and air trap of the dialyzer; as well as the temperature of the machine and dialysate). The result of all these alterations and non-physiological contacts is the activation of immunological surveillance mechanisms, which, in turn, induces an inflammatory reaction, tightly interconnected with hemostasis [26,27,28]. This is proven by the permanent identification of thrombi in the hemodialyzer circuit [29].

Among the innate immunity mechanisms triggered in hemodialyzed patients, one that we must mention is the rapid activation of C3 convertase and the initiation of the three complement pathways: classic, alternative, and lectin. This activation is proven by the elevation in concentration of C3a, C3b, C5a, C5b, along with the 70% increase in serum MAC (membrane attack complex: C5bC6C7C8C9) [30,31]. These high levels are observed at the onset of the hemodialysis sessions (first 15–30 min) and they become progressively reduced throughout the session. One of the explanations offered for this finding is complement inhibitor (H factor and clusterin) adsorption on polysulfone dialysis membranes, resulting in a temporary inability to block complement cascades [32,33].

Another way to induce inflammation during hemodialysis sessions is serum neutrophile and monocyte activatio, upon contact with the dialysis membranes. Concomitantly, a drop in neutrophile numbers and morphofunctional alterations of monocytes (mainly phenotypical) were also observed [34]. With regard to their phenotypical aspect (the existing types of membrane receptors), monocytes fall into three different categories (Mo1, Mo2, and Mo3). Mo1 monocytes have CD14 receptors for lipopolysaccharides that act as co-receptors for TLRs (Toll-like receptors). Mo2 and Mo3 have both CD14 and CD16 (FcγR III, Ig G receptor) [35]. Hemodialyzed patients have presented increased, exaggerated amounts of Mo2 and Mo3, which have both proinflammatory and atherogenic functions by binding to the endothelial cells [36].

Uremic toxins are another leukocyte activator present in hemodialyzed patients. High serum levels of urea cause the endothelial cells to increase the expression of ICAM-1, VCAM-1, and E-selectin adhesion molecules on their membranes [37]. Moreover, the uremic toxins also function as DAMPs (damage-associated molecular patterns) that bind to TLRs on the endothelial cells and circulating leukocytes and generate new mechanisms of inflammatory reaction activation and stimulation [37]. Thus, a vicious circle is created, with the continuous amplification of proinflammatory cytokine and uremic toxin release. One of the aforementioned mechanisms involves NOS3 (nitric oxide physiological endothelial synthase) blockage and NOS2 (nitric oxide inducible synthase, a strong proinflammatory agent) activation. The uremic toxins also stimulate the endothelial cells to expose tissular factor III (tissular thromboplastin) on their membranes, thus triggering the extrinsic coagulation pathway (the fastest pathway) [38].

We also observed higher levels of inflammation in patients with diabetes when compared to those without diabetes. Our results prove the significant increase of proinflammatory cytokine IL-1β, TNFα, and especially IL-6 serum levels, in all the hemodialyzed patients enrolled, even more so in the diabetics. Our result is in line with previous reports showing that chronic inflammation is involved either in the occurrence and progression of type 2 diabetes [39] (*Eur Cardiol Rev.* (2019) 14(1):50–9). Furthermore, higher levels of serum IL-6 have also been implicated in the development of type 2 diabetes [40] (*Inflammopharmacology* (2018) 26(3):685–98). However, the IL-6 elevations observed in our subjects were so important that, in order to obtain a more detailed understanding of their influences, we divided the initial group of patients into four subgroups by ratio between the observed serum IL-6 concentration and the upper limit of the normal range for IL-6 (Table 2).

IL-6 is a cytokine with roles in the regulation of the immune and nervous systems, hepatic regeneration, and metabolism control [41]. In order to elicit the aforementioned effects, IL-6 has to bind to the specific membrane receptor IL-6R. The IL-6–IL-6R complex then connects to the transmembrane glycoprotein gp130, which homodimerizes and triggers intracellular signaling [42]. Gp130 is present in all cells, while IL-6R only exists in a few types of cells (hepatocytes and some leukocytes). The cells that do not contain IL-6R cannot respond to IL-6 stimulation, and gp130 by itself does not have a measurable affinity for IL-6R. There is also another form of IL-6 receptor, the soluble IL-6R (sIL-6R), that originates from the breaking of the transmembrane receptor [42]. It has been proven that IL-6R can rupture through the intervention of bacterial and inflammatory proteases, but also as an effect of cholesterol-reducing drugs [43,44]. The sIL-6R can bind IL-6 with the same affinity as the membrane receptor (IL-6R). The IL-6–sIL-6R complex can bind to gp130 on the cellular membranes that do not express IL-6R, so it affects cells that do not directly respond to IL-6. This phenomenon is called trans-signaling [43]. An interesting and well-worth mentioning fact is that trans-signaling is proinflammatory, while classic IL-6 signaling, through the transmembrane IL-6R, is pro-regenerative and anti-inflammatory.

For the purpose of establishing the dominant IL-6 signaling mechanisms, we also determined sIL-6R serum concentrations. The significantly increased levels in the hemodialyzed diabetic patient group compared with the non-diabetic hemodialyzed patients (Figure 1) allowed for us to conclude that IL-6 induces intense proinflammatory trans-signaling in dialyzed patients, especially those with diabetes. Also, we can illustrate the existence of a vicious signaling circle in the hemodialyzed patients: the existing inflammation, triggered by the above mechanisms (especially in those with diabetes) determines the increase in IL-6 synthesis and elevated membraneIL-6R breakage, with the appearance of the soluble receptor. On the other hand, the formation of the IL-6–sIL-6R complex triggers trans-signaling on all cellular membranes, thus amplifying the already existing inflammation. Additionally, the untethering from the cell membrane of IL-6R—through which IL-6 transmits anti-inflammatory and pro-regenerative signals, followed by the trans-signaling induced by the IL-6–sIL-6R complex—generates opposing effects on the tissue regeneration capacity and triggers apoptosis. This is proven by the activation of the ADAM 17 protease, which is secondary to apoptosis initiation [45].

This is why another area of interest in our study was the evaluation of inflammation’s influence on tissue regeneration capacity in our hemodialyzed patients, especially those with diabetes. We chose neurotrophin-3 (NT-3) and β vascular endothelial growth factor (VEGFβ) as markers. The data that we have reviewed so far allows for us to say that our study is the first to approach NT-3 behavior in this category of patients.

NT-3 is the third neurotrophic factor to be discovered, after the nerve growth factor (NGF) and the brain-derived neurotrophic factor (BDNF). It is secreted in the brain, heart, and liver, as well as the pancreas and kidneys [46].

NT-3 distribution and its influence on various neuronal populations clearly differentiates NT-3 from NGF and BDNF. Thus, in order to produce its effects, NT-3 binds with high affinity to the tyrosine kinase receptor Trk C and with low affinity to the Trk A, Trk B, and p75NTR receptors. After connecting to Trk C, the receptor dimerizes and auto-phosphorylates on the tyrosine fragments in the intracellular segment. It is to these tyrosine fragments that the adaptor proteins bind and, in turn, connect with PLCγ and PI3K enzymes and the RAAS (renin–angiotensin–aldosterone system). There are also some non-functional forms of Trk C that do not have a tyrosine kinase domain, but they do compete with functional Trk C for NT-3 binding. The result would be NT-3 bioavailability and normal Trk C functionality reduction [47].

Recently, it was proven that NT-3 attenuated immune responses in cells from stroke patients and controls. The mechanism whereby human immune cells respond to NT-3 may be via Trk C receptors, whose levels are regulated by stimulation [48].

Because both chronic hemodialysis and diabetic angiopathy predispose the patient to vascular lesions, we also observed the vascular regeneration capacity of the 83 hemodialyzed individuals enrolled in this study.

Angiogenesis is a multistep process induced by many growth factors and cytokines. Two types of angiogenesis-promoting factors have been very well characterized: (1) those that act as direct angiogenic factors, such as vascular endothelial growth factors (VEGFs) and fibroblast growth factors (acidic, aFGF/FGF1, and basic bFGF/FGF2, respectively); (2) those that indirectly stimulate angiogenesis, like tumor necrosis factor α (TNF-α) and transforming growth factor β (TGFβ) [48].

Among the VEGF family, VEGFα was proven to have a strong role at the vascular system level, in both inflammation and angiogenesis [49].

Our intention was to study VEGFβ, which has less clear biological functions in hemodialyzed patients. Initially, VEGFβ was thought to have an angiogenic role, like VEGFα, because of its high sequence homology and similar receptor binding pattern. Current data show that VEGFβ actually has a more nuanced role in angiogenesis than VEGFα [50].

Under degenerative conditions, VEGFβ inhibits the apoptosis of different types of vascular cells (endothelial cells, pericytes, and smooth muscle cells), whereas in the presence of high levels of angiogenic/growth factors, it acts as an inhibitory factor, ensuring a balanced blood vessel density and tissue growth [51].

Some authors have reported that VEGFβ may potentiate angiogenesis by increasing the bioavailability of VEGFα [52], but the majority of them agree that VEGFβ cannot initiate angiogenesis or increase vascular permeability by itself [53,54].

We consider that the hemodialyzed patients included in our study, especially those with diabetes, suffer from degenerative vascular conditions, and so, VEGFβ exhibits an inhibiting effect on vascular wall cell apoptosis.

On the other hand, VEGFβ is also involved in the exchange of information between the endothelial cells and preadipocytes that occurs during both angiogenesis and adipogenesis [55].

Our results presented very low serum levels of NT-3 and VEGFβ, determined as tissular regeneration markers (Figure 6 and Figure 7). Moreover, these two parameters showed significantly altered serum concentrations in relation to the serum elevation of IL-6, as follows: for NT-3, *p* = 0.048 and for VEGFβ, *p* = 0.029. With regard to NT-3, the highest levels were obtained in the hemodialyzed diabetic group, in both sexes, for which the IL-6 serum concentration was 10–20 times higher than the upper limit of the normal range. With regard to VEGFβ, the highest serum level was obtained in the female hemodialyzed diabetic patient group, with IL-6 levels over 20 times higher than normal.

We believe that one of the incriminating causes for the general decrease in these markers is the negative trans-signaling induced by the IL-6–sIL-6R complex.

In conclusion, the idea of dividing the patients into smaller groups arose when we obtained highly elevated but very ununiform levels of IL-6 in the dialyzed patients, especially in those with diabetes. When we divided the patients into the aforementioned groups, a fundamental discovery was made: only when the IL-6 serum levels were 10–20 times over the normal value (group 10–20) did we observe maximal NT-3 concentrations (nervous and tissular regeneration). In other words, only when the serum concentrations of IL-6 were significantly increased could this cytokine signal through its fixed membrane receptors, which promote regeneration. When IL-6 levels did not exceed the threshold of 10 times the normal value, the cytokine acted upon the soluble receptors, inducing trans-signaling and blocking regeneration (as mentioned earlier in this article). These are crucial aspects that could not have been deduced without subdividing the patients into smaller groups. Similar behavior was observed in VEGFβ as well: the highest VEGFβ levels were obtained when the IL-6 concentration was over 20 times greater than the normal value. These conditions need to be met in order for VEGFβ to inhibit endothelial cell apoptosis.

While innate immunity, with the inflammatory reaction as its central event, is exacerbated in patients undergoing hemodialysis, especially diabetic ones, the acquired immunity is functionally and numerically deficient with regard to both T (LT) and B lymphocytes (LB) [56].

With regard to LTs, it has been determined that naïve LTs, helper LTs that stimulate the humoral immune response (LTh), and regulatory LTs that inhibit the immune response (LTreg) are reduced, while memory LTs are elevated [57]. One of the explanations for the decrease in naïve LTs is the reduction in bone marrow production and LT maturation, release from the thymus, and the increase in apoptosis [58]. Also, the frequent administration of recombinant erythropoietin (rhuEPO) for the correction of end-stage renal disease-induced anemia is considered to be one of the causes of LT apoptosis, especially CD4^+^ LTs [59]. It is already known that one single hemodialysis session reduces the activation capacity of mitogen-stimulated naïve CD4^+^ LTs, as well as their entrance into the first phase of their cellular cycle [60].

The same type of behavior was described for the LBs: the number of naïve LBs decreases, while memory LBs are increased [61]. One of the causes of the B cells’ shortened life cycle is the reduction in informational exchange between them and the LTh. This exchange presupposes the formation of contacts between the constitutive LB receptor CD40 and its ligand, CD40L, which appears on the membrane of activated LTh. In hemodialyzed patients, a drop in CD40 receptors was observed, because they come loose from the LB membrane and accumulate, in their soluble form (CD40s), in biological fluids, including the blood. In its soluble form, CD40 antagonizes the interaction between CD40 and CD40L [62]. The behavioral pattern of CD40 is similar to that of soluble IL-6R.

In diabetic patients, all innate and acquired immunity alterations are profoundly amplified, mainly from the endothelial level. In physiological conditions, insulin activates signaling through the PI-3K (phosphoinositide-3 kinase)/Akt pathway in endothelial cells. The result is nitric oxide (NO) synthesis, through eNOS (endothelial nitric oxide enzyme) activation. In type 2 diabetes mellitus, because of increased insulin resistance and physiological PI-3/Akt pathway deficiency, the MAPK pathway predominates, with a proatherogenic effect. Secondary to the PI-3/Akt pathway deficiency, a NO deficiency also appears, as initially proven in obese mice models [63].

Because TNFα has a very well-documented role in multiple inflammatory pathologies (innate immunity), our study aimed to evaluate cellular immunity alterations in hemodialyzed patients, including diabetic ones, by looking at TNFβ (lymphotoxin α) serum levels.

TNFα and TNFβ were isolated in 1984 from macrophages and lymphocytes [64]. Today, we know that TNFβ is expressed by a variety of cells, including T cells, B cells, and natural killer (NK) cells, with an important cytotoxic effect [65].

Both cytokines bind to the same receptor [66]. The amino acid sequence of TNFβ shows 35% identity and 50% homology to TNFα and exhibits further structural similarity in tertiary and quaternary structure, indicating similar biological activity [67].

TNFβ is secreted and, like TNFα, binds with high affinity to TNF receptors 1 and 2 (TNFR-1 and TNFR-2) on activated LTs and LBs, a phenomenon that is followed by the binding of another TNF form, TNF C. Thus, on these cellular membranes, a heterotrimer is formed by TNFβ, TNF C, and the TNFR receptor [68].

It is believed that TNFα, TNFβ, and TNF C act as an integrated signaling system [69].

Our study showed the highest TNFβ levels were registered in the female patients with the highest IL-6 increases (IL 10–20 and IL 21). This enables us to regard IL-6 as a cellular immune response stimulator, when in high concentrations, even though it is a specific marker for innate immunity. It is probable that this stimulation of TNFβ synthesis and release is not a product of trans-signaling, but of classical signaling through the intact IL-6R.

We did not obtain significant differences in TNFα, NGAL, and IL-10 concentrations between the two groups of hemodialyzed patients, nor any significant correlations between TNFα and TNFβ.

## 6. Conclusions

Although our recently obtained results represent only a primary data processing, they describe the way in which diabetes mellitus influences the chronic inflammation specific to hemodialyzed patients. Our study is also the first to approach serum NT-3 behavior as a marker of tissue regeneration in hemodialyzed patients and the influence inflammation has on this behavior. Moreover, we also obtained interesting information about VEGFβ and TNFβ in this category of patients, especially since most of the already existing data refer to VEGFα and TNFα.

The fact that this is the first study to attest the directorial role of IL-6 on immunological functions and tissue regeneration capacity in diabetic hemodialyzed patients opens a new avenue in the attempt to improve the clinical state and evolution of these patients. There still remains one unattained target, though: conducting IL-6 to signal only through its fixed membrane receptors (with anti-inflammatory and pro-regenerative effects) and blocking trans-signaling through soluble IL-6 receptors, sIL-6R (with a proinflammatory and anti-regenerative effect).

However, our study has several limitations.

The results presented in this paper are the product of a primary processing of data, without the normalization of the initial values in the database. Our study is of a cross-sectional design; thus, it limits our observations to a single point in time and does not enable us to discuss any potential clinical outcomes. Even so, our findings can constitute an important alarm signal, drawing attention to the multiple influences that diabetes mellitus has on the immunity and tissue regeneration capacity of hemodialyzed patients.

## Figures and Tables

**Figure 1 medsci-12-00031-f001:**
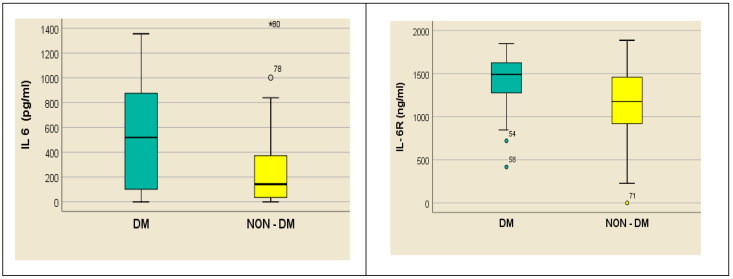
Median, outliers, and first and third quartiles of IL-6 and, respectively, IL-6R in dialyzed diabetics (DM) vs. dialyzed non-diabetics (NON-DM). For each subgroup the outliers of IL 6, or IL-6R respectively, are represented by a different dot (*, O) and the ID-number of the patient.

**Figure 2 medsci-12-00031-f002:**
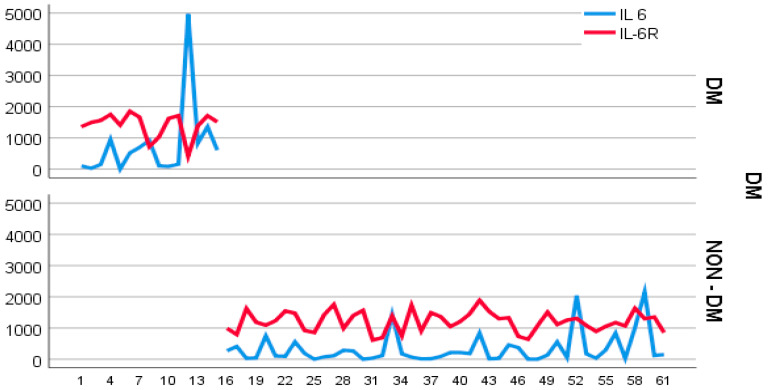
Value distribution of serum IL-6 and IL-6R in study group.

**Figure 3 medsci-12-00031-f003:**
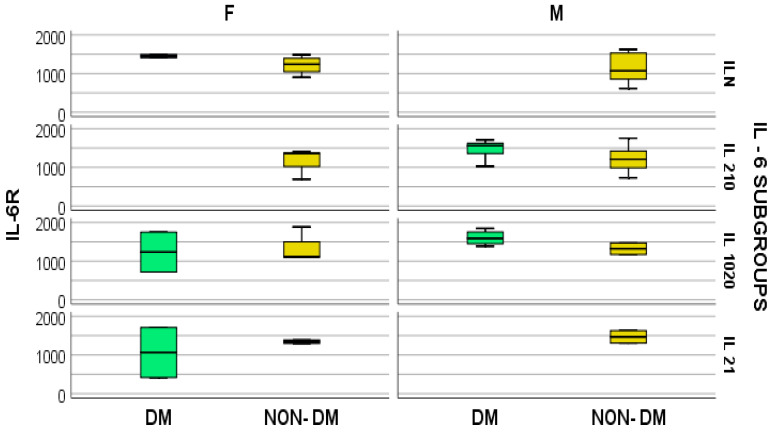
Box and whisker chart representing the dataset of IL-6R in subgroups differentiated according to the presence of diabetes, IL-6 category, and gender (DM = diabetes mellitus; F = female; M = male).

**Figure 4 medsci-12-00031-f004:**
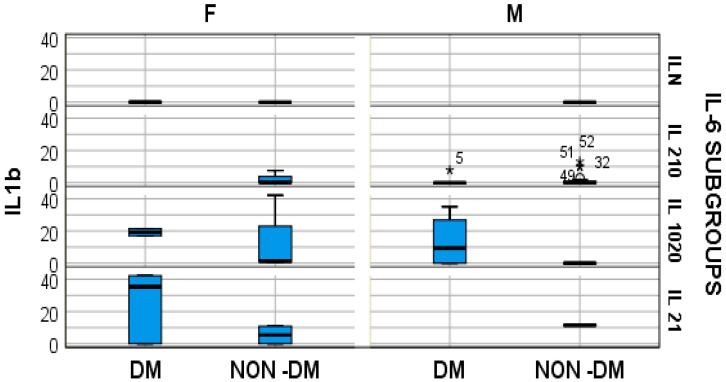
Box and whisker chart representing the dataset of IL-1β in subgroups differentiated according to the presence of diabetes, IL-6 category, and gender (DM = diabetes mellitus; F = female; M = male). For the “IL 210” IL-6 subgroup, for the male gender, the outliers of IL 1β, in the DM and NON-DM groups, respectively, are represented by “*” and the ID-number of the patients, in order to provide a better image of IL 1β distribution in this subgroup.

**Figure 5 medsci-12-00031-f005:**
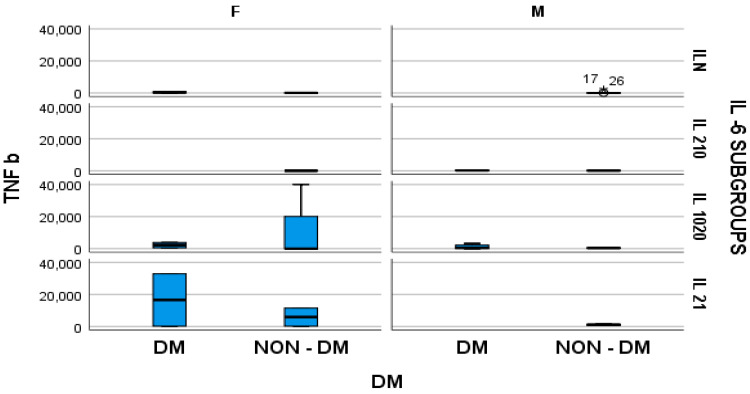
Box and whisker chart representing the dataset of TNF-β in subgroups differentiated according to the presence of diabetes, IL-6 category, and gender (DM = diabetes mellitus; F = female; M = male). For the “IL N” IL-6 subgroup, for the male gender, in the NON-DM group, the outliers of TNFβ are represented by “*” and the ID-number of the patients.

**Figure 6 medsci-12-00031-f006:**
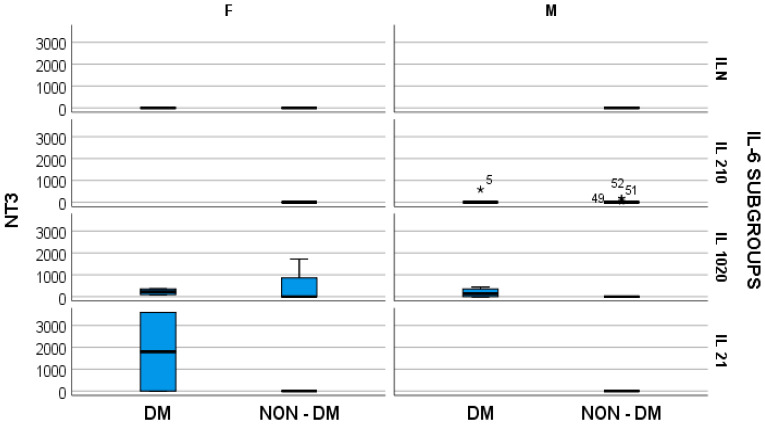
Box and whisker chart representing the dataset of NT-3 in subgroups differentiated according to the presence of diabetes, IL-6 category, and gender (DM = diabetes mellitus; F = female; M = male). For the “IL 210” IL-6 subgroup, for the male gender, the outliers of NT3, in the DM, and NON-DM groups, respectively, are represented by “*” and the ID-number of the patients.

**Figure 7 medsci-12-00031-f007:**
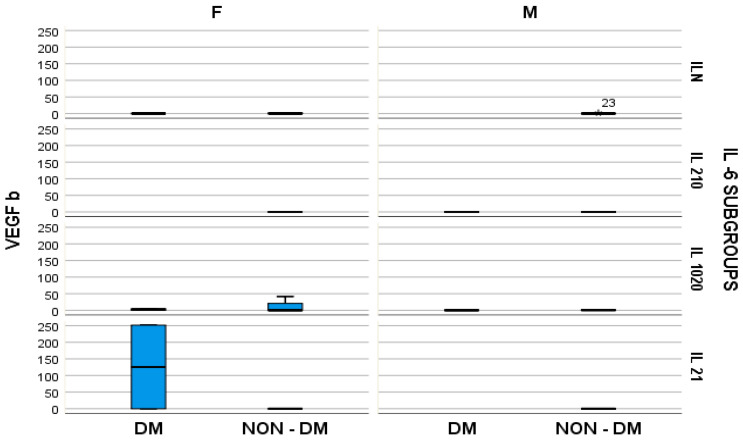
Box and whisker chart representing the dataset of VEGF-β in subgroups differentiated according to the presence of diabetes, IL-6 category, and gender (DM = diabetes mellitus; F = female; M = male). For the “IL N” IL-6 subgroup, an outlier of VEGF-β, for the male gender, in the NON-DM group is highlighted by “*” and the ID-number of the patient.

**Figure 8 medsci-12-00031-f008:**
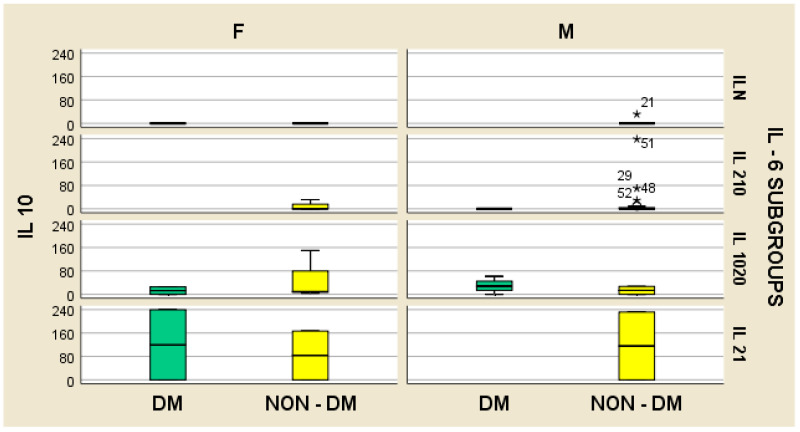
Box and whisker chart representing the dataset of IL-10 in subgroups differentiated according to the presence of diabetes, IL-6 category, and gender (DM = diabetes mellitus; F = female; M = male). The outliers of IL 10 are highlighted by “*” and the ID-number of the patient, in the NON-DM group, for the male gender, in the “IL N” IL-6 subgroup and in the “IL 210” IL-6 subgroup respectively.

**Figure 9 medsci-12-00031-f009:**
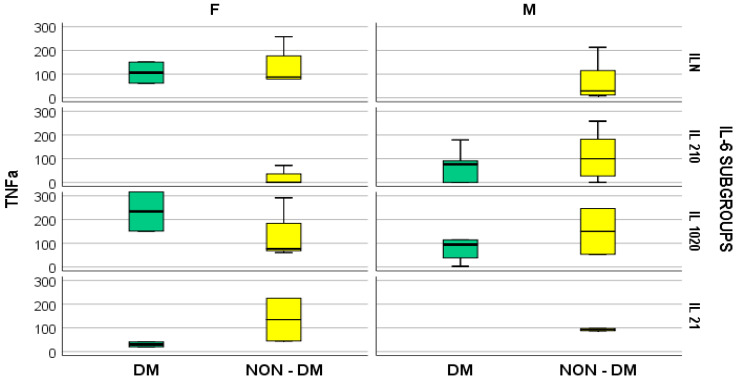
Box and whisker chart representing the dataset of TNF-α in subgroups differentiated according to the presence of diabetes, IL-6 category, and gender (DM = diabetes mellitus; F = female; M = male).

**Figure 10 medsci-12-00031-f010:**
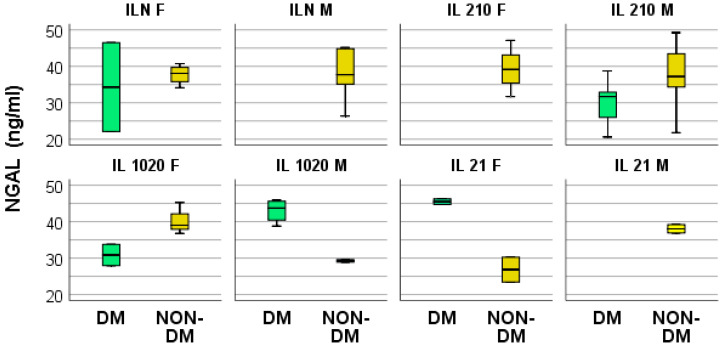
Box and whisker chart representing the dataset of NGAL in subgroups differentiated according to the presence of diabetes, IL-6 category, and gender (DM = diabetes mellitus; F = female; M = male).

**Figure 11 medsci-12-00031-f011:**
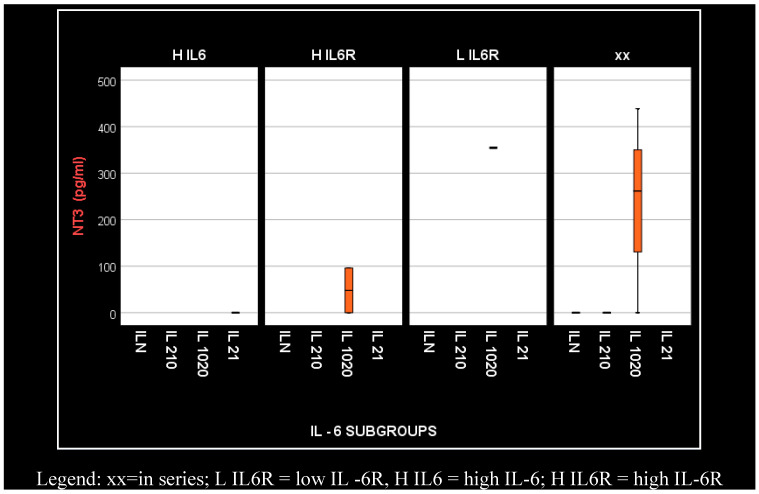
NT-3 behavior in relation to the level of IL-6 and IL-6R concentration elevation in patients with diabetes.

**Table 1 medsci-12-00031-t001:** Patient characteristics.

Subjects on Hemodialysis (n)	Age (Years)	Gender (M/F)	CKD Duration (Years)	Time on Hemodialysis (Years)
NON DM (62)	61.5 ± 15.1	33/28	8.26 ± 6.99 years.	5.29 ± 4.71
DM (21)	63 ± 11.8	9 M, 12 F	5.23 ± 4.34 years.	2.53 ± 2.69

Values are mean ± SD; M = male; F = female; CKD = chronic kidney disease.

**Table 2 medsci-12-00031-t002:** Subject distribution as range of individual values of serum IL-6 an IL-6R concentrations.

Value Range	IL-6 (pg/mL)		IL-6R (ng/mL)	
	DM (n)	NON-DM (n)	DM (n)	NON-DM (n)
<50	2	13	0	1
50–100	1	6	0	0
100–500	4	19	1	2
500–1000	7	11	3	17
≥1000	7	13	17	42

## Data Availability

The authors consent to data sharing in a publicly accessible repository.

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
