# Peer review of "Interleukin-6 as a Director of Immunological Events and Tissue Regenerative Capacity in Hemodialyzed Diabetes Patients"

_medsci, 2024, doi:10.3390/medsci12020031_

Round 1

Reviewer 1 Report

Comments and Suggestions for Authors

The aim of the study have to be expressed more precisely.

Fig.1 does not represent correlation.

I suggest including table representing concentrations of parameters both in DM and non-DM patients

Differences with p=0.05 or p=0.046 cannot be taken as significant.

The idea of dividing patients into smaller groups is not clear. 

Comments on the Quality of English Language

The paper needs extensive correction in terms of used english.

Author Response

Please see attachement.

Reviewer 2 Report

Comments and Suggestions for Authors

Comments.

1. There are some sentences in the manuscript which are not clear or need to be rewritten. (line 25, 29-31,…).

2. Author should clearly state the specific objectives and hypotheses of the study in the introduction part. Also, make a background of the study. This will provide readers with a more focused understanding of what the study aims to achieve.

3. Author should make a table for “Patients and Methods” part.

4. The figures need to be re-drawn as many of them appear damaged or incomplete.

5. Elaborate on the rationale behind the selection of specific biomarkers (IL-6, IL-6R, NT-3, VEGF-β). Explain why these markers were chosen and how they relate to inflammation and tissue regeneration in hemodialysis patients.

6. Expand the literature review to include more recent studies on inflammation and regenerative markers in hemodialysis patients. This will provide a broader context for the study and highlight its contribution to the field.

7. Author should mention the figure number in the written part of the manuscript, where they were explained.

8. Again check all the references in entire manuscript. For ex- [9,10,11] should be [9-11].  

Comments on the Quality of English Language

Extensive English editing is required.   

Author Response

Please see attachement.

Reviewer 3 Report

Comments and Suggestions for Authors

The authors investigated the relation ship between diabetes mellitus and chronic inflammation in hemodialyzed patients, with special interest to VEGF beta and TNF beta.

Comments on the Quality of English Language

Moderate English corections are needed

Author Response

Please see attachement.

Reviewer 4 Report

Comments and Suggestions for Authors

The authors investigated the immunological pattern and tissue regeneration capacity in two groups of hemodialysis patients: diabetics and non-diabetics. The researchers measured serum markers for inflammation (IL-6, IL-1β, TNF-α, sIL-6R, NGAL, IL-10), cellular immune response (TNF-β), and tissue regeneration capacity (NT-3, VEGF-β). The results showed significant increases in IL-6 and sIL-6R levels in both groups, especially in the diabetics. High IL-6 levels were found to significantly influence the behavior of IL-1β, TNF-β, NT-3, VEGF-β, and IL-10. The researchers concluded that IL-6 generates proinflammatory trans-signaling through sIL-6R, leading to reduced NT-3 and VEGF-β levels, indicating an anti-regenerative effect.

Comments

1. The study included only 21 diabetic hemodialysis patients and 62 non-diabetic hemodialysis patients. A larger sample size would increase the statistical power and allow for more robust conclusions.

2. While the study investigated several markers, it could have included additional markers for inflammation, cellular immune response, and tissue regeneration to provide a more comprehensive understanding of the immunological and regenerative processes.

3.  The study was cross-sectional, providing a snapshot of the patients' conditions.  

4. The study did not investigate the association between the immunological and regenerative markers and clinical outcomes, such as cardiovascular events, infections, or mortality.   

5. The study did not adequately address potential confounding factors, such as age, gender, duration of hemodialysis, and comorbidities, which could influence the results.    

6. The study could have provided a more in-depth discussion of the potential clinical implications of the findings, such as the development of targeted therapies or the identification of high-risk patients who may benefit from closer monitoring or interventions.

Round 2

Reviewer 1 Report

Comments and Suggestions for Authors

I am gratefull for all changes made by authors. I do not have further remarks.

Reviewer 2 Report

Comments and Suggestions for Authors

The authors addressed all my concerns. Therefore, I recommend this manuscript for publication. 

Comments on the Quality of English Language

Minor English editing is required before publication.